# Multi-LoRA Composition for Image Generation

**Ming Zhong**[1]                                                     *mingz5@illinois.edu*

**Yelong Shen**[2]                                                   *yelong.shen@microsoft.com*

**Shuohang Wang**[2]                                             *Shuohang.Wang@microsoft.com*

**Yadong Lu**[2]                                                     *yadonglu@microsoft.com*

**Yizhu Jiao**[1]                                                     *yizhuj2@illinois.edu*

**Siru Ouyang**[1]                                                   *siruo2@illinois.edu*

**Donghan Yu**[2]                                                   *donghanyu@microsoft.com*

**Jiawei Han**[1]                                                     *hanj@illinois.edu*

**Weizhu Chen**[2]                                                   *wzchen@microsoft.com*

[1] *University of Illinois Urbana-Champaign,* [2] *Microsoft Corporation*

**Reviewed on OpenReview:** *https://openreview.net/forum?id=25FTODqhVZ*

## Abstract

Low-Rank Adaptation (LoRA) is extensively utilized in text-to-image models for the accurate rendition of specific elements like distinct characters or unique styles in generated images. Nonetheless, existing methods face challenges in effectively composing multiple LoRAs, especially as the number of LoRAs to be integrated grows, thus hindering the creation of complex imagery. In this paper, we study multi-LoRA composition through a decoding-centric perspective. We present two training-free methods: LoRA Switch, which alternates between different LoRAs at each denoising step, and LoRA Composite, which simultaneously incorporates all LoRAs to guide more cohesive image synthesis. To evaluate the proposed approaches, we establish `ComposLoRA`, a new comprehensive testbed as part of this research. It features a diverse range of LoRA categories with 480 composition sets. Utilizing an evaluation framework based on GPT-4V, our findings demonstrate a clear improvement in performance with our methods over the prevalent baseline, particularly evident when increasing the number of LoRAs in a composition. The code, benchmarks, LoRA weights, and all evaluation details are available on our project website.

## 1 Introduction

In the dynamic realm of generative text-to-image models (Ho et al., 2020; Rombach et al., 2022; Saharia et al., 2022; Ramesh et al., 2022; Ruiz et al., 2023; Sohn et al., 2023), the integration of Low-Rank Adaptation (LoRA) (Hu et al., 2022) stands out for its ability to fine-tune image synthesis with remarkable precision and minimal computational load. LoRA excels by specializing in one element — such as a specific character, a particular clothing, a unique style, or other distinct visual aspects — and being trained to produce diverse and accurate renditions of this element in generated images. For instance, users could customize their LoRA models to generate various images of themselves, achieving an array of personalized and realistic representations.

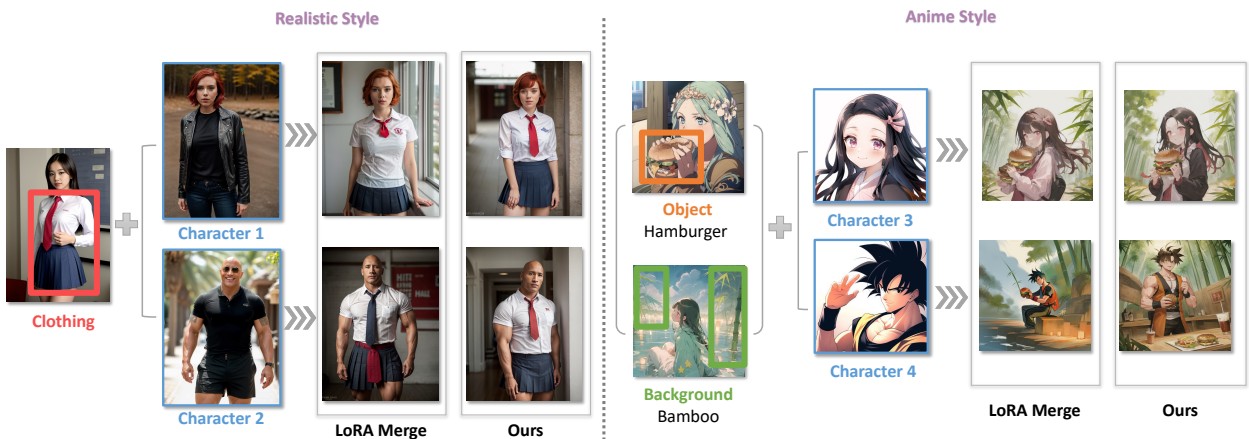

Figure 1: Multi-LoRA composition techniques effectively blend different elements such as characters, clothing, and objects into a cohesive image. Unlike the conventional LoRA Merge approach (Ryu, 2023), which can lead to detail loss and image distortion as more LoRAs are added, our methods retain the accuracy of each element and the overall image quality.

The application of LoRA not only showcases its adaptability and precision in image generation but also opens new avenues in customized digital content creation, revolutionizing how users interact with and utilize generative text-to-image models for creating tailored visual content.

However, an image typically embodies a mosaic of various elements, making **compositionality** key to controllable image generation (Tenenbaum, 2018; Huang et al., 2023b). In pursuit of this, the strategy of composing multiple LoRAs, each focused on a distinct element, emerges as a feasible approach for advanced customization. This technique enables the digitization of complex scenes, such as virtual try-ons, merging users with clothing in a realistic fashion, or urban landscapes where users interact with meticulously designed city elements. Prior investigations into multi-LoRA compositions have explored the context of pre-trained language models (Zhang et al., 2023a; Huang et al., 2023a) or stable diffusion models (Ryu, 2023; Shah et al., 2023). These studies aim to merge multiple LoRA models to synthesize a new LoRA model by training coefficient matrices (Huang et al., 2023a; Shah et al., 2023; Wu et al., 2024) or through the direct addition or subtraction of LoRA weights (Ryu, 2023; Zhang et al., 2023a). Nevertheless, these approaches centered on weight manipulation could destabilize the merging process as the number of LoRAs grows (Huang et al., 2023a) and also overlook the interaction between LoRA models and base models. This oversight becomes particularly critical in diffusion models, which depend on sequential denoising steps for image generation. Ignoring the interplay between LoRAs and these steps can result in misalignments in the generative process, as shown in Figure 1, where a merged LoRA model fails to preserve the full complexity of all desired elements, leading to distorted or unrealistic images.

In this paper, we delve into multi-LoRA composition from a decoding-centric perspective, keeping all LoRA weights intact. We present two **training-free** approaches that utilize either one or all LoRAs at each decoding step to facilitate compositional image synthesis. Our first approach, LORA SWITCH, operates by selectively activating a single LoRA during each denoising step, with a rotation among multiple LoRAs throughout the generation process. For instance, in a virtual try-on scenario, LORA SWITCH alternates between a character LoRA and a clothing LoRA at successive denoising steps, thereby ensuring that each element is rendered with precision and clarity. In parallel, we propose LORA COMPOSITE, a technique that draws inspiration from classifier-free guidance (Ho & Salimans, 2022). It involves calculating unconditional and conditional score estimates derived from each respective LoRA at every denoising step. These scores are then averaged to provide balanced guidance for image generation, ensuring a comprehensive incorporation of all elements. Furthermore, by bypassing the manipulation on the weight matrix but directly influencing the diffusion process, both methods allow for the integration of any number of LoRAs and overcome the limitations of recent studies that typically merge only two LoRAs (Shah et al., 2023).

Experimentally, we introduce `ComposLoRA`, the first testbed specifically designed for LoRA-based composable image generation. This testbed features an extensive array of six LoRA categories, spanning two distinct visual styles: reality and anime. Our evaluation includes 480 diverse composition sets, each incorporating a varying number of LoRAs to comprehensively evaluate the efficacy of each proposed method. Given the lack of standardized automatic metrics for this novel task, we propose to employ GPT-4V (OpenAI, 2023a;b) as an evaluator, assessing both the quality of the images and the effectiveness of the compositions. Our empirical findings consistently demonstrate that both LoRA SWITCH and LoRA COMPOSITE substantially outperform the prevalent LoRA merging approach, particularly noticeable as the number of LoRAs in a composition increases. To further validate our results, we also conduct human evaluations, which reinforce our conclusions and affirm the efficacy of our automated evaluation framework. In addition, we provide a detailed analysis of the applicable scenarios for each method, as well as discuss the potential bias of using GPT-4V as an evaluator.

To summarize, our key contributions are threefold:

- We introduce the first investigation of multi-LoRA composition from a decoding-centric perspective, proposing LoRA SWITCH and LoRA COMPOSITE. Our methods overcome existing constraints on the number of LoRAs that can be integrated, offering enhanced flexibility and improved quality in composable image generation.

- Our work establishes `ComposLoRA`, a comprehensive testbed tailored to this research area, featuring six varied categories of LoRAs and 480 composition sets. Addressing the absence of standardized metrics, we present an evaluator built upon GPT-4V, setting a new benchmark for assessing both image quality and compositional efficacy.

- Through extensive automatic and human evaluations, our findings reveal the superior performance of the proposed methods compared to the prevalent LoRA merging approach. Additionally, we provide an in-depth analysis of different multi-composition methods and evaluation frameworks.

## 2 Related Work

### 2.1 Composable Text-to-Image Generation

Composable image generation, a key aspect of digital content customization, involves creating images that adhere to a set of pre-defined specifications (Liu et al., 2023). Existing research in this domain primarily focuses on the following approaches: enhancing compositionality with scene graphs or layouts (Johnson et al., 2018; Yang et al., 2022; Gafni et al., 2022), modifying the generative process of diffusion models to align with the underlying specifications (Feng et al., 2023; Huang et al., 2023c;b), multi-concept customization (Kumari et al., 2023; Han et al., 2023; Gu et al., 2023; Kwon et al., 2024; Kong et al., 2024), or composing a series of independent models that enforce desired constraints (Du et al., 2020; Liu et al., 2021; Nie et al., 2021; Liu et al., 2022; Li et al., 2023; Du et al., 2023).

However, these methods typically operate at the *concept level*, where generative models excel in creating images based on broader categories or general concepts. For example, a model might be prompted to generate an image of "a woman wearing a dress", and can adeptly accommodate variations in the textual description, such as changing the color of the dress. Yet, they struggle to accurately render specific, user-defined elements, like lesser-known characters or unique dress styles. Another line of work that can compose user-defined objects into images (Huang et al., 2023c; Ruiz et al., 2023). However, these methods require extensive fine-tuning and do not perform well on multiple objects. Therefore, we introduce *learning-free instance-level* composition approaches utilizing LoRA, enabling the precise assembly of user-specified elements in image generation.

### 2.2 LoRA-based Manipulations

Leveraging large language models (LLMs) or diffusion models as the base model, recent research aims to manipulate LoRA weights to achieve a range of objectives: element composition in image generation (Ryu, 2023; Shah et al., 2023), enhancing or diminishing certain capabilities in LLMs (Zhang et al., 2023a; Huang

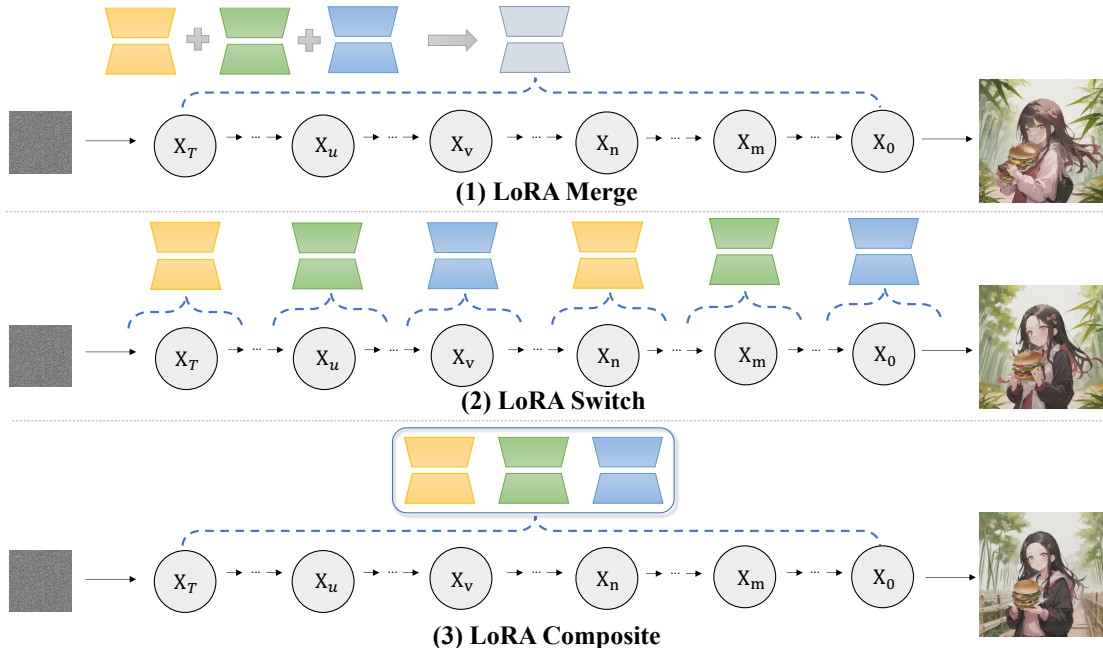

Figure 2: Overview of three multi-LoRA composition techniques, where each colored LoRA represents a distinct element. The prevalent approach, LoRA Merge, linearly merges multiple LoRAs into a single one. In contrast, our methods concentrate on the denoising process: LoRA Switch cycles through different LoRAs during the denoising, while LoRA Composite involves all LoRAs working together as the guidance throughout the generation process.

et al., 2023a), incorporating world knowledge (Dou et al., 2023), and transferring parametric knowledge from larger teacher models to smaller student models (Zhong et al., 2023). Regarding LoRA composition techniques, both LoRAHub (Huang et al., 2023a) and ZipLoRA (Shah et al., 2023) employ few-shot demonstrations to learn coefficient matrices for merging LoRAs, enabling the fusion of multiple LoRAs into a singular new LoRA. On the other hand, LoRA Merge (Ryu, 2023; Zhang et al., 2023a) introduces addition and negation operators to merge LoRA weights through arithmetic operations.

Nevertheless, these weight-based methods often lead to instability in the merging process as the number of LoRAs increases (Huang et al., 2023a). They also fail to account for the interactive dynamics when applying the LoRA model in conjunction with the base model. To address these issues, our study explores a new perspective: instead of altering the weights of LoRAs, we maintain all LoRA weights intact and focus on the interactions between LoRAs and the underlying generative process.

## 3 Method

In this section, we begin with an overview of essential concepts for understanding multi-LoRA composition, followed by detailed descriptions of our proposed methods.

### 3.1 Preliminary

**Diffusion Models.** Diffusion models (Sohl-Dickstein et al., 2015; Ho et al., 2020; Dhariwal & Nichol, 2021; Song et al., 2021; Nichol et al., 2022) represent a class of generative models adept at crafting data samples from Gaussian noise through a sequential denoising process. They build upon a sequence of denoising autoencoders that estimate the score of a data distribution (Hyvärinen, 2005). Given an image $x$, the encoder $\mathcal{E}$ is used to map $x$ into a latent space, thus yielding an encoded latent $z = \mathcal{E}(x)$. The diffusion process introduces noise to $z$, resulting in latent representation $z_t$ with different noise levels over timestep $t \in \mathcal{T}$.

The diffusion model $\epsilon_\theta$ with learnable parameters $\theta$ is trained to predict the noise added to the noisy latent $z_t$ given text instruction conditioning $c_T$. Typically, a mean-squared error loss function is utilized as the denoising objective:

$$L = \mathbb{E}_{\mathcal{E}(x), \epsilon \sim \mathcal{N}(0,1), t} \left[ ||\epsilon - \epsilon_\theta(z_t, t, c_T)||_2^2 \right], \tag{1}$$

where $\epsilon$ is the additive Gaussian noise. In this paper, we investigate multi-LoRA composition based on diffusion models, which is consistent with the settings of previous studies on LoRA merging (Ryu, 2023; Shah et al., 2023).

**Classifier-Free Guidance.** In diffusion-based generative modeling, classifier-free guidance (Ho & Salimans, 2022) balances the trade-off between the diversity and quality of the generated images, particularly in scenarios where the model is conditioned on classes or textual descriptions. For the text-to-image task, it operates by directing the probability mass towards outcomes where the implicit classifier $p_\theta(c|\mathbf{z}_t)$ predicts a high likelihood for the textual conditioning $c$. This necessitates the diffusion models to undergo a joint training paradigm for both conditional and unconditional denoising. Subsequently, during inference, the guidance scale $s \geq 1$ is used to adjust the score function $\tilde{e}_\theta(\mathbf{z}_t, c)$ by moving it closer to the conditional estimation $e_\theta(\mathbf{z}_t, c)$ and further from the unconditional estimation $e_\theta(\mathbf{z}_t)$, enhancing the conditioning effect on the generated images, as formalized in the following expression:

$$\tilde{e}_\theta(\mathbf{z}_t, c) = e_\theta(\mathbf{z}_t) + s \cdot (e_\theta(\mathbf{z}_t, c) - e_\theta(\mathbf{z}_t)). \tag{2}$$

**LoRA Merge.** Low-Rank Adaptation (LoRA) approach (Hu et al., 2022) enhances parameter efficiency by freezing the pre-trained weight matrices and integrating additional trainable low-rank matrices within the neural network. This method is founded on the observation that pre-trained models exhibit low "intrinsic dimension" (Aghajanyan et al., 2021). Concretely, for a weight matrix $W \in \mathbb{R}^{n \times m}$ in the diffusion model $\epsilon_\theta$, the introduction of a LoRA module involves updating $W$ to $W'$, defined as $W' = W + BA$. Here, $B \in \mathbb{R}^{n \times r}$ and $A \in \mathbb{R}^{r \times m}$ are matrices of a low-rank factor $r$, satisfying $r \ll \min(n, m)$. The concept of LoRA Merge (Ryu, 2023) is realized by linearly combining multiple LoRAs to synthesize a unified LoRA, subsequently plugged into the diffusion model. Formally, when introducing $k$ distinct LoRAs, the consequent updated matrix $W'$ in $\epsilon_\theta$ is given by:

$$W' = W + \sum_{i=1}^{k} w_i \times B_i A_i, \tag{3}$$

where $i$ denotes the index of the $i$-th LoRA, and $w_i$ is a scalar weight, typically a hyperparameter determined through empirical tuning. LoRA Merge has emerged as a dominant approach for presenting multiple elements cohesively in an image, offering a straightforward baseline for various applications. However, merging too many LoRAs at once can destabilize the merging process (Huang et al., 2023a), and it completely overlooks the interaction with the diffusion model during the generative process, resulting in the deformation of the hamburger and fingers in Figure 2.

### 3.2 Multi-LoRA Composition through a Decoding-Centric Perspective

To address the above issues, we base our approach on the denoising process and investigate how to perform composition while maintaining the LoRA weights unchanged. This is specifically divided into two perspectives: in each denoising step, either activate only one LoRA or engage all LoRAs to guide the generation.

**LoRA Switch (LoRA-s).** To explore activating a single LoRA in each denoising step, we present LORA SWITCH. This method introduces a dynamic adaptation mechanism within diffusion models by sequentially activating individual LoRAs at designated intervals throughout the generation process. As illustrated in Figure 2, each LoRA is represented by a unique color corresponding to a specific element, with only one LoRA engaged per denoising step.

With a set of $k$ LoRAs, the methodology initiates with a prearranged sequence of permutations; in the example of the Figure, the sequence progresses from yellow to green to blue LoRAs. Starting from the first LoRA, the model transitions to the subsequent LoRA every $\tau$ step. This rotation persists, allowing each LoRA to be applied in turn after $k\tau$ steps, thereby endowing each element to contribute repeatedly to the image generation. The active LoRA at each denoising timestep $t$, ranging from 1 to the total number of steps required, is determined by the following equations:

$$
\begin{aligned}
i &= \lfloor ((t-1) \bmod (k\tau))/\tau \rfloor + 1, \\
W'_t &= W + w_i \times B_i A_i.
\end{aligned}
\tag{4}
$$

In this formula, $i$ indicates the index of the currently active LoRA, iterating from 1 to $k$. The floor function $\lfloor \cdot \rfloor$ guarantees the integer value of $i$ is appropriately computed for $t$. The resulting weight matrix $W'_t$ is updated to reflect the contribution from the active LoRA. By selectively enabling one LoRA at a time, LORA SWITCH ensures focused attention to the details pertinent to the current element, thus preserving the integrity and quality of the generated image throughout the process.

**LoRA Composite (LoRA-c).** To explore incorporating all LoRAs at each timestep without merging weight matrices, we propose LORA COMPOSITE (LORA-C), an approach grounded in the Classifier-Free Guidance paradigm. Previous research has primarily focused on modifying CFG to enable diffusion models to emphasize textual concepts (Liu et al., 2022; Du et al., 2023; Sohn et al., 2023). In contrast, our method extends this by enabling CFG to condition on LoRAs, facilitating the generation of images that reflect specific elements or instances rather than abstract concepts. LORA-C involves calculating both unconditional and conditional score estimates for each LoRA individually at every denoising step. By aggregating these scores, the technique ensures balanced guidance throughout the image generation process, facilitating the cohesive integration of all elements represented by different LoRAs.

Formally, with $k$ LoRAs in place, let $\theta'_i$ denote the parameters of the diffusion model $e_\theta$ after incorporating the $i$-th LoRA. The collective guidance $\tilde{e}(\mathbf{z}_t, c)$ based on textual condition $c$ is derived by aggregating the scores from each LoRA, as depicted in the equation below:

$$
\tilde{e}(\mathbf{z}_t, c) = \frac{1}{k} \sum_{i=1}^{k} w_i \times \left[ e_{\theta'_i}(\mathbf{z}_t) + s \cdot (e_{\theta'_i}(\mathbf{z}_t, c) - e_{\theta'_i}(\mathbf{z}_t)) \right].
\tag{5}
$$

Here, $w_i$ is a scalar weight allocated to each LoRA, intended to adjust the influence of the $i$-th LoRA. In this paper, we set $w_i$ to 1, giving each LoRA equal importance. LORA-C assures that every LoRA contributes effectively at each stage of the denoising process, addressing the potential issues of robustness and detail preservation that are commonly associated with merging LoRAs.

Overall, we are the first to adopt a decoding-centric perspective in multi-LoRA composition, steering clear of the instability inherent in weight manipulation on LoRAs. Our study introduces two training-free methods for activating either one or all LoRAs at each denoising step, with their comparative analysis presented in Sections §4.2 and §4.3.1.

# 4 Experiments

## 4.1 Experimental Setup

**ComposLoRA Testbed.** Due to the absence of standardized benchmarks and automated evaluation metrics, existing studies involving evaluation for composable image generation lean heavily on quantitative analysis (Huang et al., 2023b; Wang et al., 2023) and human effort (Shah et al., 2023), which also limits the advancements of multi-LoRA composition. To bridge this gap, we introduce a comprehensive testbed ComposLoRA designed to facilitate comparative analysis of various composition approaches. This testbed builds upon a collection of public LoRAs[1], which are extensively shared and recognized as essential plug-in modules in this field. The selection of LoRAs for this testbed adheres to the following criteria:

---

[1]Collected from https://civitai.com/.

- Each LoRA should be robustly trained, ensuring it can accurately replicate the specific elements it represents when integrated independently;

- The elements represented by the LoRAs should cover a diverse range of categories and demonstrate adaptability across different image styles;

- When composed, LoRAs from different categories should be compatible, preventing any conflicts in the resulting image composition.

Consequently, we curate two unique subsets of LoRAs representing realistic and anime styles. Each subset comprises a variety of elements: 3 characters, 2 types of clothing, 2 styles, 2 backgrounds, and 2 objects, culminating in a total of 22 LoRAs in `ComposLoRA`. In constructing composition sets, we strictly follow a crucial principle: each set must include one character LoRA and avoid duplication of element categories to prevent conflicts. Thus, the `ComposLoRA` evaluation incorporates a total of 480 distinct composition sets. This includes 48 sets comprising 2 LoRAs, 144 sets with 3 LoRAs, 192 sets featuring 4 LoRAs, and 96 sets containing 5 LoRAs. Key features for each LoRA are manually annotated and serve dual purposes: they act as input prompts for the text-to-image models to generate images, and also provide reference points for subsequent evaluations using GPT-4V. Detailed descriptions of each LoRA can be found in Table 9 in the Appendix.

**Comparative Evaluation with GPT-4V.** While existing metrics can calculate the alignment between text and images (Hessel et al., 2021; Ku et al., 2023), they fall short in assessing the intricacies of specific elements within an image and the quality of their composition. Recently, multimodal large language models like GPT-4V (OpenAI, 2023a;b) have significant progress and promise in various multimodal tasks, underscoring their potential in evaluating image generation tasks (Lin et al., 2023; Zhang et al., 2023b). In our study, we leverage GPT-4V's capabilities to serve as an evaluator for composable image generation.

Table 1: Comparative evaluation with GPT-4V. The evaluation prompt and result are in a simplified version.

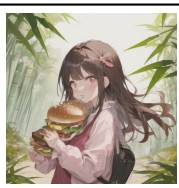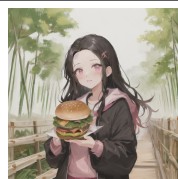

**Evaluation Prompt**
I need assistance in comparatively evaluating two text-to-image models based on their ability to compose different elements into a single image. The key elements are:

1. Character: kamado nezuko, black hair, pink eyes, forehead
2. Background: bamboolight, outdoors, bamboo
3. Object: two-handed burger, holding a huge burger

Please help me rate based on composition and image quality:

[*Evaluation criteria and format requirements*]

**Evaluation Results from GPT-4V**
For Image 1:

Composition Quality:
- The character's hair color is not black as specified (**-1 point**).
- The burger shows a deformed connection which detracts from the quality of the object's depiction (**-3 points**).
- The character's fingers are deformed (**-1 point**).

Image Quality:
- Deformities are present in the character's fingers and the burger, which significantly impacts the quality. (**-3 points**).
...

Scores:
Image 1: Composition Quality: **5/10**, Image Quality: **7/10**
Image 2: Composition Quality: **10/10**, Image Quality: **10/10**

Specifically, we employ a comparative evaluation method, utilizing GPT-4V to rate generated images across two dimensions: composition quality and image quality. We utilize a 0 to 10 scoring scale, with higher scores indicating superior quality. GPT-4V is provided with a prompt that includes the essential features of the elements to be composed, the criteria for scoring in the two dimensions, and the format for the expected output. The complete evaluation prompts and results are available in Tables 7 and 8 in Appendix. This experimental setup allows us to compare the efficacy of each of the two proposed methods against the LoRA MERGE approach. Additionally, we examine how GPT-4V-based scoring aligns with human judgment in Section §4.2 and explore the potential biases of using it as an evaluator in Section §4.3.3.

**Implementation Details.** For our experiments, we employ stable-diffusion-v1.5 (Rombach et al., 2022) as the backbone model. We utilize two specific checkpoints for our experiments: "Realistic_Vision_V5.1"[2] for realistic images and "Counterfeit-V2.5"[3] for anime images, each fine-tuned to their respective styles. In the realistic style subset, we configure the model with 100 denoising steps, a guidance scale $s$ of 7, and set the image size to 1024x768, optimizing for superior image quality. For the anime style subset, the settings differ slightly with 200 denoising steps, a guidance scale $s$ of 10, and an image size of 512x512. The DPM-Solver++ (Lu et al., 2022a;b) is used as the scheduler in the generation process. The weight scale $w$ is consistently set at 0.8 for composing LoRAs within `ComposLoRA`. For the LoRA Switch approach, we apply a cycle with $\tau$ set to 5, meaning every 5 denoising steps activate the next LoRA in the sequence: character, clothing, style, background, then object. Since the proposed methods do not require additional training, all experiments are conducted on a single A6000 GPU. To ensure the reliability of our experimental results, we conduct image generation using three random seeds. All reported results in this paper represent the average evaluation scores across these three runs.

## 4.2 Results on `ComposLoRA`

**GPT-4V-based Evaluation.** We first present the comparative evaluation results using GPT-4V. This evaluation involves scoring the performance of LoRA-s versus LoRA MERGE, and LoRA-c versus LoRA MERGE across two dimensions, as well as determining the winner based on these scores. Specific scores and win rates are illustrated in Figure 3, leading to several key observations:

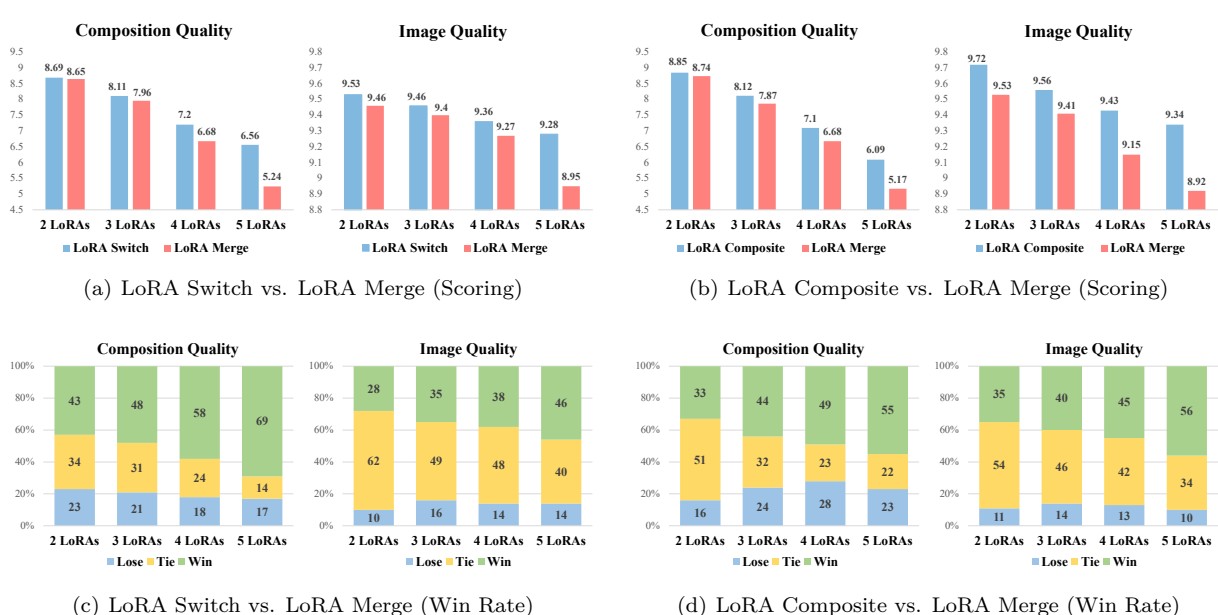

(a) LoRA Switch vs. LoRA Merge (Scoring)

(b) LoRA Composite vs. LoRA Merge (Scoring)

(c) LoRA Switch vs. LoRA Merge (Win Rate)

(d) LoRA Composite vs. LoRA Merge (Win Rate)

Figure 3: Results of comparative evaluation on `ComposLoRA` using GPT-4V.

- *Our proposed method consistently outperforms LoRA Merge across all configurations and in both dimensions, with the margin of superiority increasing as the number of LoRAs grows.* For instance, as shown in Figure 3(a), the score advantage of LoRA SWITCH escalates from 0.04 with 2 LoRAs to 1.32 with 5 LoRAs. This trend aligns with the win rate observed in Figure 3(c), where the win rate approaches 70% when composing 5 LoRAs.

- *LoRA-S shows superior performance in composition quality, whereas LoRA-C excels in image quality.* In scenarios involving 5 LoRAs and using LoRA MERGE as a baseline, the win rate of LoRA-s in

---

[2]https://huggingface.co/SG161222/Realistic_Vision_V5.1_noVAE.
[3]https://huggingface.co/gsdf/Counterfeit-V2.5.

composition quality surpasses that of LoRA-C by 14% (69% vs. 55%). Conversely, for image quality, LoRA-C's win rate is 10% higher than that of LoRA-S (56% vs. 46%).

- *The task of compositional image generation remains highly challenging, especially as the number of elements to be composed increases.* According to GPT-4V's scoring, the average score for composing 2 LoRAs is above 8.5, but it sharply declines to around 6 for compositions involving 5 LoRAs. Hence, despite the considerable improvements our methods offer, there is still substantial room for further research in the field of compositional image generation.

**Human Evaluation.** To complement our results, we conduct a human evaluation to assess the effectiveness of different methods and validate the efficacy of the evaluators.

Two graduate students rate 120 images on compositional and image quality using a 1-5 Likert scale: 1 signifies complete failure, 2-4 represents significant, moderate, and minor issues, respectively, while 5 denotes perfect execution. To ensure consistency, the annotators initially pilot-score 20 images to standardize their understanding of the criteria. The results, summarized in the upper section of Table 2, align with GPT-4V's findings, confirming our methods outperform LoRA Merge — with LoRA Switch excelling in composition and LoRA Composite in image quality.

Table 2: Human evaluation results and Pearson correlation between different metrics and human judgment.

| *Human Evaluation* | | |
|---|---|---|
| | **Composition** | **Image Quality** |
| LoRA Merge | 3.14 | 2.94 |
| LoRA Switch | **3.91** | 4.15 |
| LoRA Composite | 3.78 | **4.35** |
| *Correlations with Human Judgments* | | |
| | **Composition** | **Image Quality** |
| CLIPScore | -0.006 | 0.083 |
| Ours | **0.454** | **0.457** |

Furthermore, we analyze the Pearson correlations between human evaluations and scores derived from GPT-4V and CLIPScore (Hessel et al., 2021), with results presented in the lower section of Table 2. This comparison reveals that CLIPScore's evaluations fall short in assessing specific compositional and quality aspects due to its inability to discern the nuanced features of each element. In contrast, the evaluator we adopt shows substantially higher correlations with human judgments, affirming the validity of our evaluation framework.

### 4.3 Analysis

To enhance our understanding of the proposed methods, we further investigate the following questions:

### 4.3.1 Do Specific Image Styles Favor Different Methods?

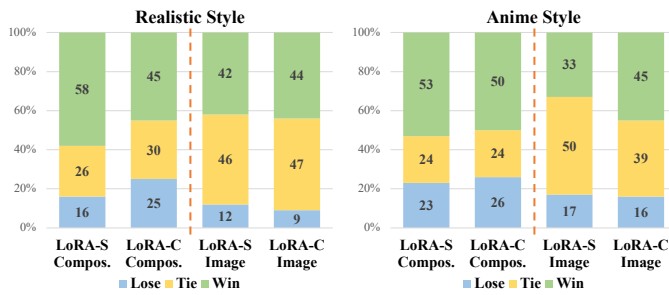

Figure 4: Analysis on image styles. In general, LoRA-S is more adept at realistic styles, while LoRA-C has better performance in anime styles.

To explore the impact of image style, we separately evaluate the performance of methods on realistic and anime-style subsets within `ComposLoRA`. The win rate results, presented in Figure 4, reveal distinct tendencies for each method.

Our observations reveal that, while LoRA-S may not excel in image quality compared to LoRA-C, it demonstrates comparable performance in this dimension within the realistic style subset, while maintaining a significant edge in composition quality. In contrast, in the anime-style subset, LoRA-C, shows a performance on par with LoRA-S in composition quality, while notably surpassing it in image quality. These findings suggest that *LoRA-S is*

*more adept at composing elements in realistic-style images, whereas LoRA-C shows a stronger performance in anime-style imagery.*

### 4.3.2 How Does the Step Size and Order of LoRA Activation Affect LoRA Switch?

To identify the optimal configuration for LORA SWITCH, we examine the influence of two crucial hyperparameters: the sequence in which LoRAs are activated and the interval between each activation. Our findings, depicted in Figure 5(a), show that overly frequent switching, such as changing LoRAs at every denoising step, leads to distortions in generated images and suboptimal performance. *The efficiency of the LoRA Switch improves progressively with increased step size, reaching peak performance at $\tau = 5$.*

Moreover, our analysis underscores that *the initial choice of LoRA in the activation sequence clearly influences overall performance, while alterations in the subsequent order have minimal impact.* Activating the character LoRA first leads to the best performance, as demonstrated in Figure 5(b). In contrast, starting with clothing, background, or object LoRAs yields results comparable to a completely randomized sequence. Notably, beginning with the style LoRA leads to a noticeable performance drop, even falling slightly below a random order. This observation underlines the critical role of prioritizing core image elements in the initial stage of the generation process to enhance both the image and compositional quality for LORA SWITCH.

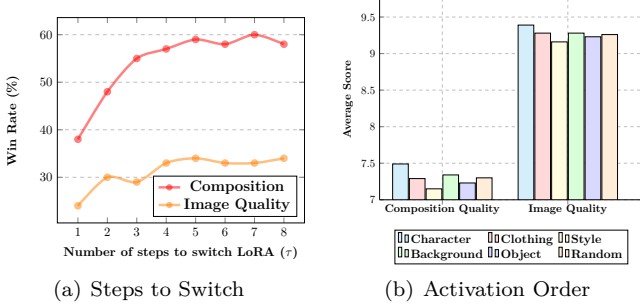

(a) Steps to Switch    (b) Activation Order

Figure 5: Analysis of the number of denoising steps to switch LoRA and the activation order for LoRA Switch. In Figure 5(b), "Character" indicates that the character LoRA is activated first, with the rest being activated randomly.

While the step size for switching LoRAs proves to be a crucial factor in achieving optimal performance in our experiments, we also explore the potential of dynamic strategies for step size adjustment throughout the denoising process. Specifically, we design and evaluate three strategies for dynamically adjusting the step size in LORA-SWITCH:

- **Incremental Strategy:** The step size gradually increases from $\tau = 3$ to $\tau = 5$ throughout the denoising process.

- **Decremental Strategy:** The step size gradually decreases from $\tau = 5$ to $\tau = 3$ as the denoising process progresses.

- **Warm-up Strategy:** During the initial 50% of the denoising process, the step size increases from $\tau = 3$ to $\tau = 5$ and remains constant at $\tau = 5$ for the remaining denoising steps.

Table 3: Performance comparison of dynamic strategies for LORA-S.

|  | $\tau = 3$ | $\tau = 4$ | $\tau = 5$ | $\tau = 6$ | Incremental ($3 \to 5$) | Decremental ($5 \to 3$) | Warm-up |
|---|---|---|---|---|---|---|---|
| Composition Quality | 55 | 57 | 59 | 58 | 57 | 54 | 58 |
| Image Quality | 29 | 33 | 34 | 33 | 33 | 31 | 34 |

Table 3 shows the results of these strategies. Neither the Incremental nor the Warm-up strategies significantly improve performance compared to using a fixed step size. The Decremental strategy, on the other hand, results in comparatively worse performance, highlighting that switching LoRAs too frequently in the latter

stages of denoising is detrimental to image quality. The fixed step size of $\tau = 5$ yields the best performance. Consequently, we adopt this fixed step size in our experiments.

### 4.3.3 Does GPT-4V Exhibit Bias as an Evaluator?

While GPT-4V has demonstrated utility in evaluating various image generation tasks (Lin et al., 2023; Zhang et al., 2023b), our analysis uncovers a notable positional bias in its comparative evaluations. We investigate this potential bias by swapping the positions of images generated by different methods before inputting them to GPT-4V, and the results are illustrated in Figure 6.

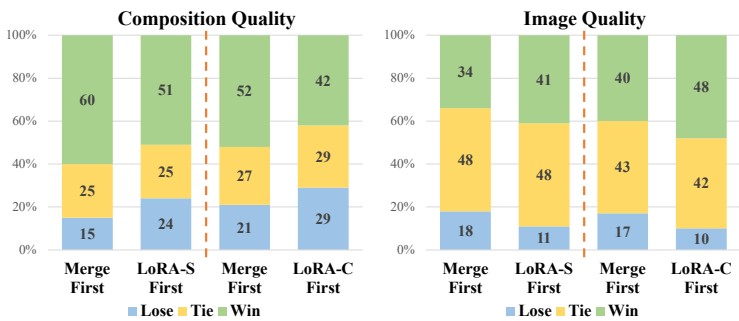

In the comparison of LoRA-s versus LoRA MERGE, when the image generated by MERGE is presented first ("Merge First"), the win rate for LoRA-s in composition quality stands at 60%. However, this win rate declines to 51% when LoRA-s's image is the first input ("LoRA-S First"). Similarly, LoRA-c's win rate decreases from 52% to 42%, suggesting that GPT-4V tends to favor the second image input in terms of composition quality. Intriguingly, the opposite trend is observed in image quality, where the second image tends to receive a higher score. *These results indicate a significant positional bias in GPT-4V's evaluation, varying with the dimension and the position of the images.* To mitigate this bias in our study, the comparative evaluation

Figure 6: Positional bias analysis for GPT-4V-based evaluation. In each subfigure, the left side of the orange line compares LoRA-s with MERGE, and the right side contrasts LoRA-c with MERGE. "Merge First" indicates that the image produced by LoRA MERGE is the first image input during the comparative evaluation.

results reported in this paper are averaged across both input orders.

### 4.4 More Visual Examples

To demonstrate the effectiveness of our methods in composing varying numbers of LoRAs and under different image styles, we provide additional visual examples in Figures 7 – 10.

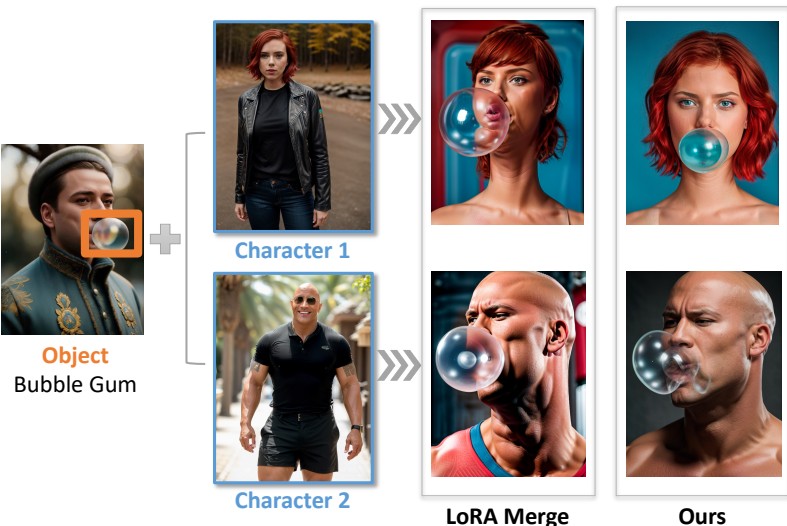

Figure 7: Case study on composing 2 LoRAs in the realistic style.

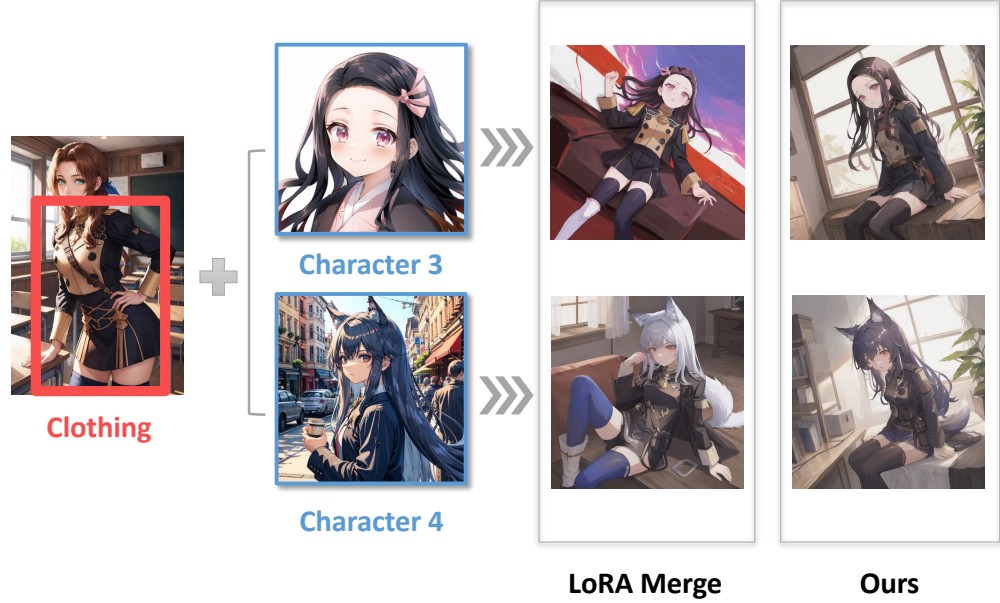

Figure 8: Case study on composing 2 LoRAs in the anime style.

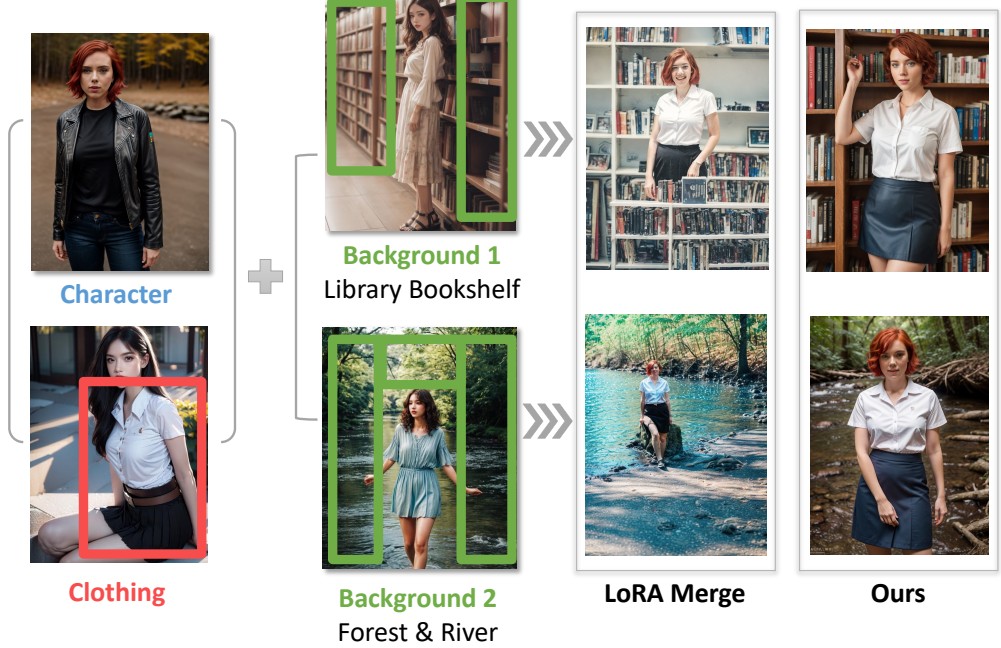

Figure 9: Case study on composing 3 LoRAs in the realistic style.

## 5 Further Discussions

### 5.1 Limitations

Based on our experiments, the primary limitation of the proposed methods is the efficiency issue with LoRA-C. This method can introduce $(k-1)\times$ additional computational cost, where $k$ is the number of

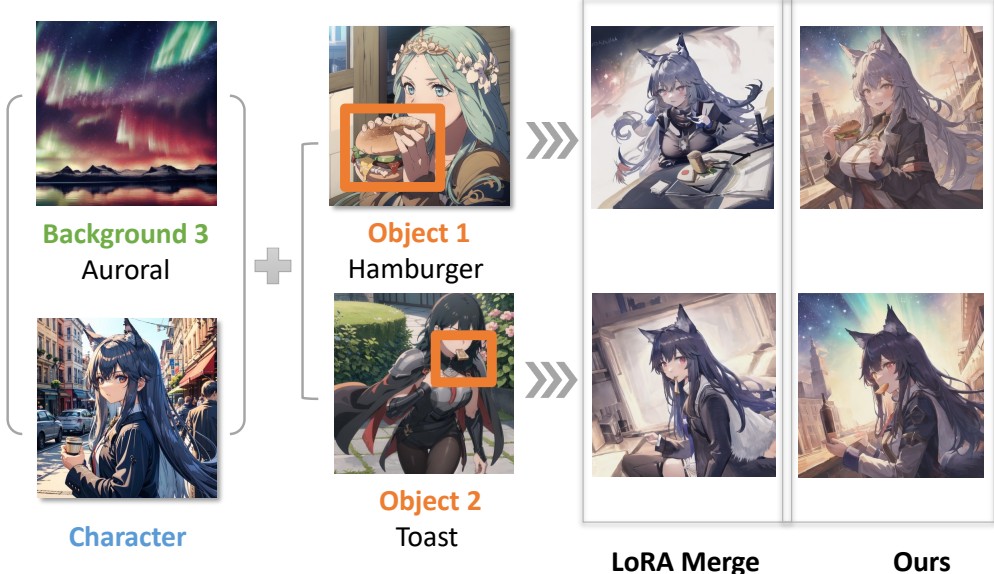

Figure 10: Case study on composing 3 LoRAs in the anime style.

merged LoRAs. This is due to LORA-C merging each LoRA with the base model to calculate scores, which are then averaged. The inherent design of LoRA prevents the pre-computation of the base model. To address this, we propose two potential solutions: 1) Integrating advanced techniques with fewer denoising steps, and 2) a combination of LORA-S and LORA-C.

### 5.1.1 Integration of LCM and LCM-LoRA

Recently, several algorithms have been developed that require only 2-8 denoising steps to generate high-quality images. We select LCM (Luo et al., 2023a) and LCM-LoRA (Luo et al., 2023b) to test if our approaches can integrate smoothly with these algorithms.

Table 4: Results for integrating LCM and LCM-LoRA.

| Integrated Technique | Method | Win (%) | Tie (%) | Lose (%) | Inference Steps |
|---|---|---|---|---|---|
| None | LORA SWITCH | 43 | 34 | 23 | 200 |
| | LORA COMPOSITE | 33 | 51 | 16 | 200 |
| LCM | LORA SWITCH | 45 | 36 | 19 | 4 |
| | LORA COMPOSITE | 39 | 43 | 18 | 4 |
| LCM-LoRA | LORA SWITCH | 42 | 39 | 19 | 8 |
| | LORA COMPOSITE | 44 | 41 | 15 | 8 |

First, we conduct experiments on 2 LoRAs using the same setup as described in the main text, with results shown in Table 4. Remarkably, our methods not only outperform the baseline but also show even greater advantages when integrated with these inference-accelerating techniques. This adaptation significantly reduces the required number of denoising steps to 4-8, effectively addressing the increased computational demands of LORA-C. Consequently, the generation times across all three methods are now comparably short, taking only a few seconds.

To further explore the potential of integrating our method with fewer denoising steps, we conduct additional experiments using LCM-LoRA with an increased number of LoRAs. These experiments aim to provide deeper insights into how our approach performs as the complexity of multi-LoRA composition increases. The results, shown in Table 5, reflect the composition quality scores under this setup. For these experiments, we use 8

denoising steps for 2 and 4 LoRAs, and 9 inference steps for 3 LoRAs, switching to the next LoRA every 3 steps for LoRA-s.

Table 5: Composition quality scores for different methods when using LCM-LoRA with more LoRAs.

|  | 2 LoRAs | 3 LoRAs | 4 LoRAs |
|---|---|---|---|
| LoRA Merge | 7.52 | 5.16 | 3.49 |
| LoRA Switch | 8.08 | **6.39** | **5.08** |
| LoRA Composite | **8.13** | 6.07 | 4.53 |

Our methods exhibit a more pronounced advantage over the baseline in this fewer-steps setting. The improvement is particularly noticeable when the number of LoRAs increases, demonstrating the robustness of our approach even when integrated with models requiring fewer denoising steps. However, it is important to note that all methods experience a substantial drop in absolute scores when combined with models like LCM-LoRA that employ fewer denoising steps. For instance, the composition quality score for LoRA-c with 5 LoRAs is initially 6.56 for 200 denoising steps, but with 4 LoRAs in this setting, the score drops to 5.08. This finding suggests that despite the improved performance of our method, multi-LoRA composition remains a challenging task, especially when fewer denoising steps are used, even with the latest integration techniques.

### 5.1.2 Combination of LoRA-s and LoRA-c

To further enhance efficiency, we propose combining LoRA-s and LoRA-c. LoRA-s activates at the LoRA stage before each denoising step, while LoRA-c is applied to the CFG during the denoising process. These design principles are complementary. A practical integration method involves selecting a subset of LoRAs to activate (ranging from one to all) for each denoising step, following the LoRA-s strategy. This subset would then utilize all its LoRAs during the denoising phase, adhering to the LoRA-c strategy. For LCM and other related applications, combining LoRA-s and LoRA-c can enhance efficiency without additional modifications. For example, in a 1-step scenario, all LoRAs can be activated and applied through CFG (as per LoRA-c). In a 2-step scenario, half of the LoRAs can be activated at each step and then applied through CFG, blending LoRA-s and LoRA-c.

### 5.2 Comparison with ZipLoRA

Table 6: Comparison with ZipLoRA in two different LoRA setups.

| LoRA Setup | Methods | Composition Quality | Image Quality |
|---|---|---|---|
| Character + Style | LoRA-S | 8.80 | 8.95 |
|  | LoRA-C | 8.55 | 9.20 |
|  | ZipLoRA | **9.05** | **9.40** |
| Character + Object | LoRA-S | **8.85** | 9.05 |
|  | LoRA-C | 8.60 | **9.25** |
|  | ZipLoRA | 8.50 | 9.10 |

Although our proposed methods are training-free, we also compare them with the fine-tuning approach ZipLoRA as a baseline for reference. ZipLoRA is based on SDXL and focuses on merging two LoRAs, so we conduct our comparisons under this setup. Specifically, we randomly select publicly available SDXL LoRAs from HuggingFace and create 10 composition sets, combining character + style and character + object for the experiments. All results are compared against the LoRA Merge, and the scores are presented in Table 6.

Since ZipLoRA is specifically designed to merge subject and style LoRAs, it achieves higher scores in the character + style setup compared to our methods, likely due to the benefits of its fine-tuning process. However,

in scenarios involving two subjects, such as character + object, our methods demonstrate clear advantages despite being training-free. LoRA-S outperforms in composition quality, while LoRA-C excels in image quality. This indicates that our methods are particularly effective in handling diverse subject combinations, especially when the task involves composing two subjects rather than merging a subject with a style.

# 6 Conclusion

In this paper, we present the first exploration of multi-LoRA composition from a decoding-centric perspective by introducing LoRA-S and LoRA-C that transcend the limitations of current weight manipulation techniques. Through establishing a dedicated testbed `ComposLoRA`, we introduce scalable automated evaluation metrics utilizing GPT-4V. Our study not only highlights the superior quality achieved by our methods but also provides a new standard for evaluating LoRA-based composable image generation.

## Broader Impact Statement

Our approaches offer advancements in personalized image generation and customized digital content creation by allowing the combination of arbitrary elements. This capability can be applied to various real-world scenarios, such as virtual try-on and virtual design, leading to positive social impacts. As our method operates in the inference phase and relies solely on the composition of publicly available checkpoints (base models and LoRAs) without requiring additional training, it avoids any negative impact related to model training.

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

# A  Appendix

## A.1  Details for Comparative Evaluation using GPT-4V

See Table 7 for the full prompts and Table 8 for a case study of evaluation results.

Table 7: The full version of evaluation prompts for comparative evaluation with GPT-4V.

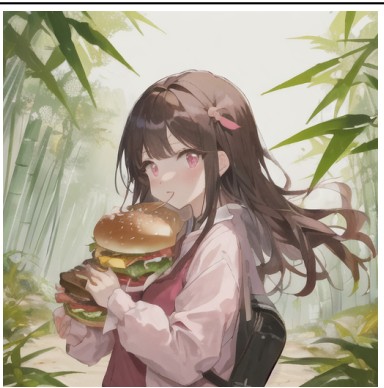 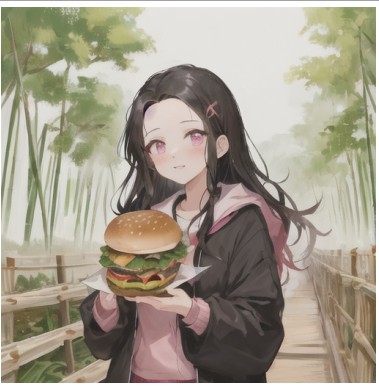

**Evaluation Prompt**

I need assistance in comparatively evaluating two text-to-image models based on their ability to compose different elements into a single image. The elements and their key features are as follows:

1. Character (Kamado Nezuko): kamado nezuko, black hair, pink eyes, forehead
2. Background (Bamboo Background): bamboolight, outdoors, bamboo
3. Object (Huge Two-Handed Burger): two-handed burger, holding a huge burger

Please help me rate both given images on the following evaluation dimensions and criteria:

Composition Quality:
   - Score on a scale of 0 to 10, in 0.5 increments, where 10 is the best and 0 is the worst.
   - Deduct 3 points if any element is missing or incorrectly depicted.
   - Deduct 1 point for each missing or incorrect feature within an element.
   - Deduct 1 point for minor inconsistencies or lack of harmony between elements.
   - Additional deductions can be made for compositions that lack coherence, creativity, or realism.

Image Quality:
   - Score on a scale of 0 to 10, in 0.5 increments, where 10 is the best and 0 is the worst.
   - Deduct 3 points for each deformity in the image (e.g., extra limbs or fingers, distorted face, incorrect proportions).
   - Deduct 2 points for noticeable issues with texture, lighting, or color.
   - Deduct 1 point for each minor flaw or imperfection.
   - Additional deductions can be made for any issues affecting the overall aesthetic or clarity of the image.

Please format the evaluation as follows:

For Image 1:
[*Explanation of evaluation*]

For Image 2:
[*Explanation of evaluation*]

Scores:
Image 1: Composition Quality: [*score*]/10, Image Quality: [*score*]/10
Image 2: Composition Quality: [*score*]/10, Image Quality: [*score*]/10

Based on the above guidelines, help me to conduct a step-by-step comparative evaluation of the given images. The scoring should follow two principles:
   1. Please evaluate critically.
   2. Try not to let the two models end in a tie on both dimensions.

Table 8: The full version of evaluation results from GPT-4V for comparative evaluation.

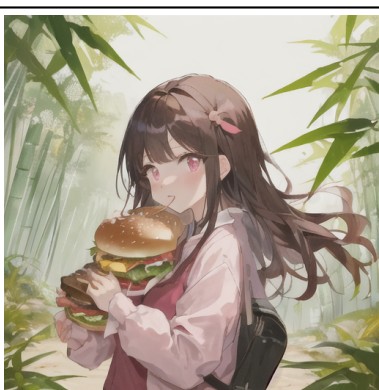 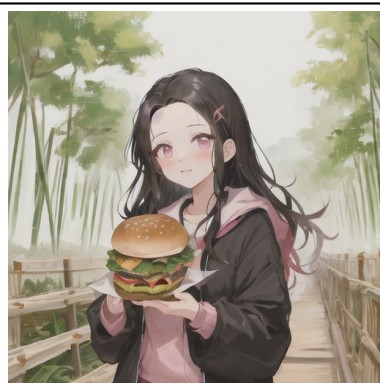

**Evaluation Results from GPT-4V**

To evaluate the images provided, I'll assess the composition quality and image quality based on the criteria outlined in your instructions. I will identify each element and its key features, and then rate the images accordingly.

For Image 1:

Composition Quality:
  - Elements: All key elements are present: character, background, and object (0 points deducted).
  - Features within the elements:
    - The character's hair color is not black as specified (**-1 point**).
    - The burger shows a deformed connection which detracts from the quality of the object's depiction (**-3 point**).
    - The character's fingers are deformed (**-1 point**).
The bamboo background is accurate, fulfilling the outdoor setting criteria. Despite this, the main issues lie with the incorrect hair color and the deformities in the burger and fingers, leading to deductions.

Image Quality:
  - The image is free from texture, lighting, or color issues (0 points deducted).
  - Deformities are present in the fingers and the burger, which significantly impacts the image's quality. (**-3 point**).

The clarity of the image is maintained, but the deformities notably affect the overall aesthetic quality.

For Image 2:

Composition Quality:
  - Elements: All key elements are present: character, background, and object (0 points deducted).
  - Features within the elements:
    - The character has black hair and pink eyes; the forehead is shown, and the burger is depicted accurately (0 points deducted).
There are no inconsistencies or lack of harmony between elements, and the composition is coherent and aesthetically pleasing.

Image Quality:
  - There are no deformities in the image (0 points deducted).
  - The texture, lighting, and color are well executed (0 points deducted).

The image maintains a high level of clarity and overall aesthetic appeal, with no visible imperfections or issues.

Scores:
Image 1: Composition Quality: **5/10**, Image Quality: **7/10**
Image 2: Composition Quality: **10/10**, Image Quality: **10/10**

In summary, Image 2 achieves perfect scores across both dimensions, demonstrating excellent composition and image quality. Image 1, while containing all the key elements, suffers in its composition due to incorrect features and deformities, and in image quality due to the noted deformities.

## A.2 Details for `ComposLoRA`

See Table 9 for the detailed descriptions of each LoRA in `ComposLoRA` Testbed.

Table 9: Detailed descriptions of each LoRA in the `ComposLoRA`.

| LoRA | Category | Trigger Words | Source |
|------|----------|---------------|--------|
| *Anime Style Subset* | | | |
| Kamado Nezuko | Character | kamado nezuko, black hair, pink eyes, forehead | Link |
| Texas the Omertosa in Arknights | Character | omertosa, 1girl, wolf ears, long hair | Link |
| Son Goku | Character | son goku, spiked hair, muscular male, wristband | Link |
| Garreg Mach Monastery Uniform | Clothing | gmuniform, blue thighhighs, long sleeves | Link |
| Zero Suit (Metroid) | Clothing | zero suit, blue gloves, high heels | Link |
| Hand-drawn Style | Style | lineart, hand-drawn style | Link |
| Chinese Ink Wash Style | Style | shuimobysim, traditional chinese ink painting | Link |
| Bamboolight Background | Background | bamboolight, outdoors, bamboo | Link |
| Auroral Background | Background | auroral, starry sky, outdoors | Link |
| Huge Two-Handed Burger | Object | two-handed burger, holding a huge burger with both hands | Link |
| Toast | Object | toast, toast in mouth | Link |
| *Realistic Style Subset* | | | |
| IU (Lee Ji Eun, Korean singer) | Character | iu1, long straight black hair, hazel eyes, diamond stud earrings | Link |
| Scarlett Johansson | Character | scarlett, short red hair, blue eyes | Link |
| The Rock (Dwayne Johnson) | Character | th3r0ck with no hair, muscular male, serious look on his face | Link |
| Thai University Uniform | Clothing | mahalaiuniform, white shirt short sleeves, black pencil skirt | Link |
| School Dress | Clothing | school uniform, white shirt, red tie, blue pleated microskirt | Link |
| Japanese Film Color Style | Style | film overlay, film grain | Link |
| Bright Style | Style | bright lighting | Link |
| Library Bookshelf Background | Background | lib_bg, library bookshelf | Link |
| Forest Background | Background | slg, river, forest | Link |
| Umbrella | Object | transparent umbrella | Link |
| Bubble Gum | Object | blow bubble gum | Link |

