# OpenReview forum: "Multi-LoRA Composition for Image Generation"
_TMLR — Accepted by TMLR_

### Review · Reviewer_1bp4 · 2024-09-18

**Summary Of Contributions:**

The paper proposed several analysis and discussions on merging multiple LoRAs for the text-to-image diffusion model generations. Specifically, they are Lora-switch/composite, which is comparable to the lora-merge baseline approaches. The proposed approach goes beyond the simple linear combination of the lora-merge approach by utilizing the CFG-inspired approaches, or simply switching between different Loras during the inference step. Utilizing GPT-4V for evaluation, their findings shows some promising results using the proposed ideas on a variety of adapter combinations.

**Audience:**

Yes

**Claims And Evidence:**

Yes

**Requested Changes:**

N/A

**Strengths And Weaknesses:**

I think this is a nice paper that proposed baseline results for composing multiple LoRAs. I enjoy reading the background sections as well as related works. The proposed approach is also illustrated well with insightful diagrams. The approaches are simple but this is a paper that discussed important baseline approaches, I think the technical correctness trumps the actual novelty in value. The use of GPT-4V is also an interesting approach for quality evaluations, but I think probably the human evaluations should be more reliable. It's also good to know that the paper will release code and this could have a bigger impact for the research community in the related field.

I couldn't find gnawing issues/weaknesses for this simple yet informative + technically sound paper (which I think suits the acceptance criteria).  My understanding is that the paper is targeting on composing more than 2 LoRAs. When there are only two LoRAs, I wonder if there are some study against some baseline approaches such as ZipLoRA.

---

> ### Author Response · Authors · 2024-10-16
> **Response to Reviewer 1bp4**
>
> We extend our thanks for the careful consideration and valuable insights you've shared. Our answer to your question is presented as follows.
>
> > **Q1: Regarding ZipLoRA baseline**
>
> Thank you for your suggestion. We agree that including fine-tuning methods as baselines provides additional insights. As ZipLoRA is based on SDXL and focuses on merging two LoRAs, we conduct comparisons under this setup. Specifically, we randomly select publicly available SDXL LoRAs from HuggingFace and create 10 composition sets combining character + style and character + object for the experiments. All results are compared against the LoRA Merge. The scores are presented below:
>
> | LoRA Setup       | Methods | Composition Quality | Image Quality |
> |----------------------|-------------|-------------------------|-------------------|
> |  | LoRA-S      | 8.80                    | 8.95              |
> |         Character + Style             | LoRA-C      | 8.55                    | 9.20              |
> |                      | ZipLoRA     | **9.05**                | **9.40**          |
> |  | LoRA-S      | **8.85**                | 9.05              |
> |          Character + Object            | LoRA-C      | 8.60                    | **9.25**          |
> |                      | ZipLoRA     | 8.50                    | 9.10              |
>
> - Since ZipLoRA is specifically designed to merge subject and style LoRAs, it achieves higher scores in the character + style setup compared to our methods, likely due to the benefits of its fine-tuning process.
> - However, in scenarios involving two subjects, such as character + object, our methods show advantages despite being training-free. This indicates that our methods are particularly effective in handling diverse subject combinations, especially when the task involves composing two subjects rather than merging a subject with a style.
>
> Thank you again for your positive feedback and support. If you have any further questions or concerns, we would be happy to address them.

---

> > ### Comment · Reviewer_1bp4 · 2024-10-28
> > **reply**
> >
> > I thank the authors for addressing my comments.

---

### Review · Reviewer_xLAv · 2024-09-19

**Summary Of Contributions:**

The authors propose two methods for merging LoRA models trained separately on multiple concepts, such as specific anime characters or styles, to generate images that encompass these concepts. One method switches between different LoRA models one by one during each denoising process (LoRA Switch), while the other draws conditional score estimates away from unconditional ones during denoising, similar to classifier-free guidance (LoRA Composite). These approaches have the advantage of not requiring additional tuning processes, unlike existing models. Moreover, to quantitatively assess the model's performance in generating images that merge up to five concepts, the authors devised a new benchmark and an automated evaluation method using GPT-4V, which demonstrates significantly improved performance compared to the baseline.

**Audience:**

No

**Claims And Evidence:**

Yes

**Requested Changes:**

- Regarding the second weakness, it would strenghthen the paper to include more recent works targeting the personalization of multiple concepts, such as Custom Diffusion [1], OMG [2], etc.
- Regarding the third weakness, presenting results from fine-tuning methods, such as ZipLoRA, could further emphasize the effectiveness of the proposed work.

[1] Multi-Concept Customization of Text-to-Image Diffusion, Kumari et al., CVPR, 2023
[2] OMG: Occlusion-friendly Personalized Multi-concept Generation in Diffusion Models, Kong et al., ECCV, 2024

**Strengths And Weaknesses:**

**Strengths**
- The proposed methods and the motivations behind them are well described in detail.
- The authors included various analysis results on the devised evaluation method, suggesting that it could be robustly utilized in future works.
- The improvements achieved by the proposed method are clearly presented in both quantitative and qualitative evaluation results.


**Weaknesses**
- The comparison was made only with a very simple baseline that sums multiple LoRA sets using empirically determined weights.
- Studies that personalize multiple concepts through modifier token tuning instead of LoRA are insufficiently addressed in related works.
- There is no analysis regarding the performance of fine-tuning methods. It is probable that those methods demonstrate significantly better quality, which could offset the advantage of the proposed work of not requiring further training.

---

> ### Author Response · Authors · 2024-10-16
> **Response to Reviewer xLAv**
>
> Thanks for the detailed review and helpful comments. In response to your questions, our answers are as follows.
>
> > **Q1: Regarding more related work**
>
> Thank you for pointing this out and for providing the helpful references! These works are indeed highly relevant to our paper, and we have added the following citations on multi-concept customization in the revised version:
>
> [1] Kumari et al., Multi-Concept Customization of Text-to-Image Diffusion. CVPR 2023.
>
> [2] Han et al., SVDiff: Compact Parameter Space for Diffusion Fine-Tuning. ICCV 2023.
>
> [3] Gu et al., Mix-of-Show: Decentralized Low-Rank Adaptation for Multi-Concept Customization of Diffusion Models. NeurIPS 2023.
>
> [4] Kwon et al., Concept Weaver: Enabling Multi-Concept Fusion in Text-to-Image Models. CVPR 2024.
>
> [5] Kong et al., OMG: Occlusion-friendly Personalized Multi-Concept Generation in Diffusion Models. ECCV 2024.
>
> Additionally, these works align with the distinction we draw in the related work section. While they focus on concept-level composition in images, our approach targets instance-level composition. Please feel free to let us know if you believe there are any other relevant studies we should consider.
>
> > **Q2: Regarding comparison with ZipLoRA**
>
> Thank you for the suggestion. We agree that including fine-tuning methods as baselines provides additional insights. Since ZipLoRA is based on SDXL and focuses on merging two LoRAs, we conduct comparisons under this setup. Specifically, we randomly select publicly available SDXL LoRAs from HuggingFace and create 10 composition sets combining character + style and character + object for the experiments. All results are compared against the LoRA Merge. The scores are presented below:
>
> | LoRA Setup       | Methods | Composition Quality | Image Quality |
> |----------------------|-------------|-------------------------|-------------------|
> |  | LoRA-S      | 8.80                    | 8.95              |
> |         Character + Style             | LoRA-C      | 8.55                    | 9.20              |
> |                      | ZipLoRA     | **9.05**                | **9.40**          |
> |  | LoRA-S      | **8.85**                | 9.05              |
> |          Character + Object            | LoRA-C      | 8.60                    | **9.25**          |
> |                      | ZipLoRA     | 8.50                    | 9.10              |
>
> - Since ZipLoRA is specifically designed to merge subject and style LoRAs, it achieves higher scores in the character + style setup compared to our methods, likely due to the benefits of its fine-tuning process.
> - However, in scenarios involving two subjects, such as character + object, our methods show advantages despite being training-free. This indicates that our methods are particularly effective in handling diverse subject combinations, especially when the task involves composing two subjects rather than merging a subject with a style.
>
> We hope these additional discussions further emphasize the effectiveness of our proposed approach.

---

### Review · Reviewer_bhh7 · 2024-10-03

**Summary Of Contributions:**

This work studied the multi-LoRA composition from a decoding-centric perspective. It proposed two training-free approaches that apply either one or all LoRAs at each decoding step to facilitate compositional image synthesis. This paper also introduced ComposLoRA, a comprehensive testbed with several public LoRAs and 480 composition sets, which can serve as benchmarks and evaluation tools for comparative analysis of various composition approaches.

**Audience:**

Yes

**Broader Impact Concerns:**

NA.

**Claims And Evidence:**

Yes

**Requested Changes:**

While the paper suggests that the LoRA switch mechanism is a form of dynamic adaptation, I would think that a truly dynamic mechanism would involve selecting which LoRA to activate at each time step in a more flexible and adaptive manner. As it stands, the switch mechanism operates each LoRA in a fixed order without real-time adaptation. Exploring more adaptive strategies for choosing which LoRA to activate at each step $t$ —potentially based on the current state values $x_t$ -- could offer further insights.

The use of 200 denoising steps is a noted limitation in terms of speed. It would be interesting to see how the proposed method performs with fewer denoising steps or if applied to other types of diffusion models, such as consistency models, which typically require fewer time steps and may improve efficiency.

For LCM and LCM-LoRA experiments, it would be helpful to specify the exact number of denoising steps used, rather than just mentioning a range of 1-8, since 1 step seems to be insufficient for the lora-Switch method. Additionally, it would be insightful to see how the performance (with just 1-8 steps) holds up when the number of LoRAs is increased, for example, to five, as this could further test the scalability and robustness of the proposed methods.

**Strengths And Weaknesses:**

Strengths: The composition in image generation is of great importance and finds applications in various domains. This paper proposes new techniques for multi-LoRA composition, demonstrated by good performance in numerical experiments.

Weaknesses: While the proposed method is intuitive, it could benefit from a few deeper insights to justify why this particular switching mechanism is effective or advantageous. For example, beyond switching loras for a fixed order, one could also consider switching in random orders or employing a more adaptive mechanism that adjusts based on the specific needs of each time step. Overall, the paper could benefit from a more in-depth discussion on the rationale behind the simple switching strategy. Additionally, as discussed in the paper, the computational efficiency is a concern, the use of 200 denoising steps is computationally expensive, and for the lora-switching method, it seems that the number of denoising steps needs to be sufficiently large to ensure effectiveness, especially when the number of loras increases.

---

> ### Author Response · Authors · 2024-10-16
> **Response to Reviewer bhh7 - Part 1**
>
> We appreciate your thorough review as well as constructive feedback, and we try to answer your questions as follows.
>
> > **Q1: Regarding adaptive strategies for LoRA-S**
>
> Thank you for your insightful comments. We fully agree that a more detailed analysis of the switching strategy helps improve the community’s understanding and supports more effective use of LoRA-S. Below, we discuss the two key aspects of the switching mechanism: activation order and step size.
>
> 1. **Activation Order:** First, we would like to clarify that our experiments are not limited to a fixed activation order. As discussed in Section 4.3.2, we explore various strategies for activating LoRAs to identify the optimal approach. The results, shown in Figure 5(b), indicate that only the first activated LoRA has a significant impact on performance. For example, activating the character LoRA first leads to notably better outcomes. In contrast, the order of the remaining LoRAs has minimal impact, with performance being almost identical across multiple random trials. We believe this is because LoRA-generated images are typically character-centered, so activating the LoRA related to the most central element first is critical for generating accurate details, while the order of the subsequent LoRAs has only a minor effect on the overall performance.
>
> 2. **Step Size for Switching:** In contrast, the step size for switching LoRAs proves to be a more crucial factor in our experiments. Figure 5(b) demonstrates that varying the step size has a significant effect on performance, leading us to adopt a fixed step size of 5 for our final setting. However, we acknowledge that a more dynamic switching strategy could offer additional insights. To explore this further, we design the following three strategies for implementing dynamic step size for LoRA-S:
>
> - `Incremental Strategy`: The step size gradually increases from τ = 3 to τ = 5 throughout the denoising process.
> - `Decremental Strategy`: The step size gradually decreases from τ = 5 to τ = 3 as the denoising process progresses.
> - `Warm-up Strategy`: In the initial 50% of the denoising process, the step size increases from τ = 3 to τ = 5 and then remains constant at τ = 5.
>
> |                          |  τ = 3  |  τ = 4  |  τ = 5  |  τ = 6  | Incremental (3 → 5) | Decremental (5 → 3) | Warm-up  |
> |--------------------------|---------|---------|---------|---------|---------------------|---------------------|----------|
> | Composition Quality   |    55   |    57   |    59   |    58   |         57          |         54          |    58    |
> | Image Quality         |    29   |    33   |    34   |    33   |         33          |         31          |    34    |
>
> - Neither Incremental nor Warm-up strategies clearly impact the results.
> - The Decremental strategy leads to comparatively worse performance, indicating that switching LoRAs too frequently in the latter stages of denoising is disadvantageous.
>
> This expanded discussion is also included in the revised paper. Hope it can address your concerns and provide further insights into the effective use of LoRA-S.
>
> > **Q2: Regrading the use of 200 denoising steps**
>
> The use of 200 denoising steps in our experiments is not because our methods require 200 steps to be effective, but rather to ensure a fair comparison with the baselines by evaluating all methods at their optimal performance. To address the concern about this, we evaluate all the approaches with 50 denoising steps (the default inference step setting in the diffusers library) and compare the composition quality between 50 and 200 steps, as shown in the following Tables:
>
> `LoRA Switch vs. LoRA Merge`:
> |                    | 2 LoRAs | 3 LoRAs | 4 LoRAs | 5 LoRAs |
> |--------------------|---------|---------|---------|---------|
> | LoRA Merge (50 steps)  |   8.53  |   7.91  |   6.71  |   5.08  |
> | LoRA Merge (200 steps) |   8.65  |   7.96  |   6.68  |   5.24  |
> | LoRA Switch (50 steps) |   **8.72**  |   8.07  |   7.12  |   6.50  |
> | LoRA Switch (200 steps) |   8.69  |   **8.11**  |   **7.20**  |   **6.56**  |
>
> `LoRA Composite vs. LoRA Merge`:
> |                    | 2 LoRAs | 3 LoRAs | 4 LoRAs | 5 LoRAs |
> |--------------------|---------|---------|---------|---------|
> | LoRA Merge (50 steps)  |   8.76  |   7.69  |   6.65  |   5.04  |
> | LoRA Merge (200 steps) |   8.74  |   7.87  |   6.68  |   5.17  |
> | LoRA Composite (50 steps) | 8.79 |   **8.16**  |   **7.15**  |   6.01  |
> | LoRA Composite (200 steps) | **8.85** |   8.12  |   7.10  |   **6.09**  |
>
> - Overall, the performance between 50 and 200 denoising steps is very close across all three methods.
> - With 50 denoising steps, our method still shows a significant advantage over the baselines.
> - In some cases, using 50 denoising steps even yields better results than 200 steps, such as in the 3 LoRAs and 4 LoRAs settings for LoRA Composite.

---

> ### Author Response · Authors · 2024-10-16
> **Response to Reviewer bhh7 - Part 2**
>
> > **Q3: Regarding experiments with fewer denoising steps (LCM and LCM-LoRA)**
>
> Thank you for highlighting this point. In the revised version of the paper, we have added a specific column in the table to clearly indicate the number of inference steps used. For LCM, the typical denoising steps are set between 2-4, and for LCM-LoRA, they are usually set between 4-8 steps. To ensure optimal performance, we use 4 denoising steps for LCM and 8 steps for LCM-LoRA in our experiments. When comparing methods, both our approach and the baselines use the same inference steps to ensure a fair comparison.
>
> We also agree that increasing the number of LoRAs under this setup could provide additional insights. Therefore, we have conducted extra experiments for LCM-LoRA with an increased number of LoRAs and included them in the revised paper. The Table below shows the composition quality scores, where 2 and 4 LoRAs use 8 denoising steps, and 3 LoRAs use 9 inference steps, switching to the next LoRA every 3 steps for LoRA-S:
>
> |                    | 2 LoRAs | 3 LoRAs | 4 LoRAs |
> |--------------------|---------|---------|---------|
> | LoRA Merge     |   7.52  |   5.16  |   3.49  |
> | LoRA Switch     |   8.08  | **6.39**| **5.08**|
> | LoRA Composite  | **8.13**|   6.07  |   4.53  |
>
> - Our method demonstrates a more significant advantage over the baselines in the fewer-steps setting.
> - However, it is important to note that all methods experience a substantial drop in absolute scores when integrating with models like LCM-LoRA that require fewer denoising steps.
> - For instance, the composition quality score for LoRA Composite with 5 LoRAs is originally 6.56 (for 200 denoising steps), but with 4 LoRAs in this setting, it drops to 5.08. This suggests that even with the latest integration techniques, multi-LoRA composition remains a highly challenging task when fewer denoising steps are used.
>
> We welcome any additional feedback if there are further concerns regarding the points discussed above.

---

### Author Response · Authors · 2024-10-16
**General Response**

Dear Reviewers,

We are deeply grateful for your insightful feedback and valuable suggestions. Your comprehensive reviews have guided us in making significant enhancements to our work. We've responded individually to each reviewer's questions. Based on your reviews, we have made thorough revisions to our manuscript, `highlighting these changes in blue` for clarity. Below is a summary of the key updates:

- Added more related studies on multi-concept customization (Page 3)
- Expanded discussions on dynamic strategies for LoRA-S (Table 3, Page 10)
- Specified inference steps in our experiments (Table 4)
- Conducted experiments with increased LoRAs using fewer steps (Table 5, Pages 11-12)
- Included comparisons with ZipLoRA (Table 6, Pages 12-13)

We sincerely thank you again for your contributions to improving our work. If there are any further concerns or queries, we are fully prepared to address them.

---

### Decision · Action_Editor_NQ31 · 2024-11-10

**Recommendation:** Accept as is

**Comment:**

As all the reviewers agreed, this paper meets the expectations of the TMLR criteria. Although the proposed method is not very novel and the improvement is somewhat marginal, I think this paper is worth discussing with TMLR's audiences.

During the author-reviewer discussion period, the authors addressed the raised concerns by the reviewers. Also, I checked that the revised paper includes the discussions provided by the authors. I think this paper is okay to be published at TMLR as is.

As a minor comment, I think that it would be great if the example figures in A.3 were highlighted in the main paper. For example, put the demo page URL in the abstract, and adding more visual examples in the experiment section (note that there is no page limit for TMLR).

**Audience:**

The proposed problem is sufficiently interesting to many audiences in this field.

**Claims And Evidence:**

This paper proposes two training-free methods (LoRA Switch and LoRA Composite) to merge multiple LoRA models separately trained on different concepts. The first method LoRA Switch selectively uses one LoRA for each denoising step, while LoRA Composite uses all LoRA weights based on estimated unconditional and conditional scores. As another contribution, this paper introduces the ComposLoRA benchmark with several public LoRA weights and 480 composition sets.

This paper claims that composing multiple LoRA weights is challenging, especially when the number of LoRAs grows. As the evidence, this paper shows that the existing method, LoRA Merge qualitatively and quantitatively performs worse than the proposed method. Specifically, I think Figure 1 and Figure 3 support the original claim well qualitatively and quantitatively.

---

> ### Author Response · Authors · 2024-11-18
> **Camera-Ready Version Submitted**
>
> Dear AE,
>
> Thank you for your thoughtful suggestions and your appreciation of our work. We’ve incorporated all the reviews and your input into the camera-ready version, which has now been uploaded. Please let us know if you have any further comments or requested revisions. Thank you again for your valuable feedback!
>
> Authors of Paper 3250